# Mayaro Virus: The State-of-the-Art for Antiviral Drug Development

**DOI:** 10.3390/v14081787

**Published:** 2022-08-16

**Authors:** Ana Paula Andreolla, Alessandra Abel Borges, Juliano Bordignon, Claudia Nunes Duarte dos Santos

**Affiliations:** 1Laboratório de Virologia Molecular, Instituto Carlos Chagas, ICC/Fiocruz, Curitiba 81350-010, PR, Brazil; 2Departamento de Biologia Celular e Molecular, Universidade Federal do Paraná, Curitiba 81530-900, PR, Brazil; 3Laboratório de Pesquisas em Virologia e Imunologia, Universidade Federal de Alagoas, Maceió 57072-900, AL, Brazil

**Keywords:** arbovirus, arthritogenic alphavirus, Mayaro virus, antivirals

## Abstract

Mayaro virus is an emerging arbovirus that causes nonspecific febrile illness or arthralgia syndromes similar to the Chikungunya virus, a virus closely related from the *Togaviridae* family. MAYV outbreaks occur more frequently in the northern and central-western states of Brazil; however, in recent years, virus circulation has been spreading to other regions. Due to the undifferentiated initial clinical symptoms between MAYV and other endemic pathogenic arboviruses with geographic overlapping, identification of patients infected by MAYV might be underreported. Additionally, the lack of specific prophylactic approaches or antiviral drugs limits the pharmacological management of patients to treat symptoms like pain and inflammation, as is the case with most pathogenic alphaviruses. In this context, this review aims to present the state-of-the-art regarding the screening and development of compounds/molecules which may present anti-MAYV activity and infection inhibition.

## 1. Introduction

Arboviruses are viral diseases transmitted by arthropod vectors (arthropod-borne virus) to vertebrate hosts during blood feeding [1,2]. Arboviruses have a wide geographic distribution, predominantly in countries with tropical and subtropical climates, such as Brazil [3]. This class of viruses is leading to a growing concern of viral dispersion and new transmission cycles due to its ability to adapt to new vectors and environmental conditions [4].

There are more than 100 species of arboviruses that cause disease in humans, classified into five families: *Peribunyaviridae*, *Flaviviridae*, *Reoviridae*, *Rhabdoviridae*, and *Togaviridae* [1,5]. The arboviruses that currently pose the major threats to public health in Brazil are Dengue (DENV), Zika (ZIKV), and Yellow Fever (YFV) of the family *Flaviviridae*, and the Chikungunya virus (CHIKV) of the family *Togaviridae* [3]. ZIKV and CHIKV were considered neglected arboviruses until the recent reemergence of both, leading to explosive epidemics and causing severe clinical symptoms [6,7]. Additionally, among neglected arboviruses, the Mayaro virus (MAYV), is endemic in some regions of Central America, the Caribbean, and South America [8]. Although MAYV is frequently detected in the northern and central-western regions of Brazil, it has been spreading to non-endemic regions in recent years, posing a risk of causing epidemics in large areas of the country [9,10,11,12,13,14]. Because most of the populations in these areas are immunologically naïve to MAYV, and considering the high potential rate of vector infestation, the scenario for large-scale epidemics seems plausible [3,15,16,17]. It is worth noting that some degree of cross-protection against MAYV by humoral response induced by a previous CHIKV infection has been observed due to the close phylogenetic and antigenic relationships between CHIKV and MAYV [18,19,20].

In general, MAYV infections in humans cause focal outbreaks characterized by nonspecific febrile illness or arthralgia syndromes [9,21,22,23,24,25]. The main symptoms include chills, fever, gastrointestinal manifestations, dizziness, itching, eye pain, myalgia, and arthralgia; they can last for months to years, thus making MAYV more debilitating than that caused by other arboviruses, whose symptoms might last approximately three to four weeks [23,26]. Severe complications like myocarditis and neurological and hemorrhagic manifestations can also occur due to MAYV infection [15,24,25,26,27,28,29].

Epidemiological studies demonstrate a growing number of confirmed cases of MAYV in endemic and non-endemic regions, both in Brazil and the Americas, and the occurrence of imported human cases in other continents like Europe [8,21,22,23,26,30,31,32,33,34]. However, due to clinical manifestations often confused with other arboviruses, such as DENV, there is a large number of underreported MAYV cases, as shown by retrospective studies, especially during other arboviruses epidemics [9,11,32,35]. Several urban vectors potentially able to transmit MAYV, like *Aedes aegypti,* might also contribute to viral spread and human infection [36]. Recent surveillance studies have demonstrated the circulation and dispersion of MAYV to previously unreported areas, as well as the detection of human infections in urban settings, evidencing the potential risk of epidemics [11,37]. If epidemics caused by MAYV eventually occur, there are no vaccines or other prophylactic methods or effective antivirals approved for human use to prevent or treat the disease [38]. The clinical approach with anti-inflammatory drugs, antipyretics, and analgesics is usually used to treat secondary symptoms, and, in risk areas, arthropod vector control is implemented [39]. Thus, there is an urgent need to search for new substances/molecules that specifically target MAYV infections.

## 2. Viral Structure and Replication Cycle

### 2.1. Viral Particle

Mayaro virus (MAYV) is a member of the family *Togaviridae*, genus *Alphavirus*, belonging to the Semliki forest serocomplex. The viral particle has an icosahedral capsid of about 70 nm in diameter and a lipid envelope derived from the host cell in which heterodimers of the transmembrane glycoproteins E1 and E2 are embedded [40]. The heterodimers are ordered in trimers on the virion surface [41]. The genetic material is composed of a single strand of positive-sense RNA of ~12 kb with two open reading frames (a 7 kb and a 4 kb) that each encodes a polyprotein, separated by a short non-coding sequence (Figure 1). From the cleavage of the non-structural polyprotein, the following proteins are found: nsP1, involved in mRNA capping; nsP2, with helicase/protease activities; nsP3, which has three recognized domains (the macrodomain, the alphavirus unique domain (AUD) and the hypervariable region) all of which are required for replication; nsP4, which is solely responsible for the RNA synthetic properties of the viral replicase complex because it contains the core RdRp domain and motifs. After processing of the structural polyprotein, there are associated proteins that result in the viral particle: C, capsid; E1, envelope glycoproteins of higher molecular mass 1; E2, envelope glycoproteins of higher molecular mass 2; E3, lower mass glycoprotein; 6K, protein that associates E1 and E2; TF, transframe protein involved in viral assembly/budding and also virulence factor [42] (Figure 1) [21,41,43,44,45,46,47,48]. Notably, the E3 protein is absent in the virion structure at 4.4 Ǻ resolution from cryo-electron microscopy analysis (cryo-EM) [41].

### 2.2. Internalization

The MAYV replication cycle and the precise function of each viral protein produced are not completely understood (Figure 2). The putative protein functions are inferred from other alphaviruses. The binding of the virions to the cell receptor is primarily mediated by the E2 protein [49]. The target cell receptors that mediate the entry of the virus are also not fully known; however, the adhesion molecule Mxra8 (also known as DICAM, ASP3, or limitrin, a member of the immunoglobulin superfamily) was recently described as a receptor for MAYV and other arthritogenic alphaviruses, like CHIKV, Ross River, O’nyong nyong, and Barmah Forest virus [46,50]. However, for encephalitic alphaviruses, such as VEEV and EEV, this receptor, when blocked, does not seem to affect the infection [50]. The uptake of the viral particle is closely linked to the lipid rafts of cell membranes, where cholesterol is the main component [49]. Endocytosis occurs dependent and/or independent of clathrin (caveolar via), the former pathway most frequently used by MAYV [49]. Once endocytosed, the clathrin layer (when present) is quickly removed from the vesicle, and the virus is delivered to the initial compartment of the endosome. The presence of ATP-dependent proton pumps is the critical characteristic of the endosome, which causes acidification of the vesicle during the maturation stages. This pH change is fundamental to viral RNA delivery to the cell cytoplasm, as it will be the initial event of intermembrane fusion (vesicle and viral envelope) [49,51].

The events of alphavirus membrane fusion are comprised of three steps: (1) dissociation of E1 from E2; (2) E1 insertion into the target membrane; and (3) the formation of E1 homotrimers [51,52]. Intermembrane fusion starts from the acidification of the endosome, which will promote the dissociation of the two envelope proteins (E1 and E2). These proteins are associated in metastable conformation in the form of heterodimers, and upon dissociation, the E1 fusion loop is exposed and inserted into the target membrane of the vesicle. In a pH-independent interaction, the stem region of the E1 complex folds, distorting the target membrane through the cooperative action of several trimers, providing the necessary energy to mediate membrane fusion. Fusion proceeds through a hemifusion step where the two outer leaflets fuse. The final stable post-fusion E1 homotrimer is formed, located on the same side as the fusion loops, and transmembrane domains are anchored. Concomitantly, the E1 homotrimer refolds, resulting in complete fusion and thus forming the fusion pore, which will allow the exit of the viral nucleocapsid (NC) from the endosome to the cytoplasm of the cell [26,44,49,51].

### 2.3. Replication

Once the NC is in the cytoplasm of the cell, its disassembly is facilitated by interactions with free ribosomes [53,54,55]. Cellular ribosomes recognize the CAP region and the 5′-AUG initiator codon of the viral RNA in the cytoplasm, leading to the translation of 49S (genomic) RNA and its processing [53,54,56]. This entire process occurs within bulb-like structures present in the plasma membrane, known as spherules (Figure 2), and the structure of these membranes as a type I cytopathic vacuole (CPV-I) [57,58,59]. After recognition of the CAP region, the non-structural genes are recognized and translated, giving rise to polyproteins P123 (precursor of nsP1, nsP2, and nsP3) in higher amounts and P1234 (P123 and nsP4) in lower amounts. This phenomenon is due to the presence of a stop codon at the junction of nsP3 and nsP4 [60,61]. In both situations, the P123 polyprotein is processed sequentially, individualizing each protein into mature species. Through its proteolytic function, mature nsP2 will catalyze cleavages on P1234 releasing RNA-dependent RNA polymerase (nsP4), which forms the replication complex (P123+nsP4) when associated with P123 [62], involved in the synthesis of complementary negative (−) RNA (Figure 2) [62,63,64,65,66]. In addition, nsP2 will promote the shutdown of transcription mediated by RNA polymerase II of the host. This process was evidenced through the interaction of nsP2 with subunit A of RNA polymerase II (Rpb1) and subunit 2 of transcription initiation factor IIE (TFIIE2), suggesting a shutdown of cellular transcription [67]. In addition to the pivotal role of nsP2, the release of nsP1 is also reported as essential for creating a replication complex by associating with nsP4 (nsP1 + P23 + nsP4), which will use the (−) RNA as a template to synthesize new positive strands of genomic RNA [61,62,64]. However, the recent study by Gorchakov et al. (2008) [68] with the SINV indicated that the P123/nsP4 complex was able to synthesize (+) RNAs without the need for cleaving nsP1 from the P123 protein [69]. This process is still not fully understood with regard to MAYV. In addition to the transcription of (+) RNA, a subgenomic mRNA (26S transcript) is also transcribed, which will be translated into a structural polyprotein and give rise to structural proteins after cleavage (Figure 2) [70].

From the structural polyprotein precursor, the protein C is the first structural protein generated through its processing by the action of the autoproteolytic serine protease to encapsulate the newly synthesized RNA molecules, originating new NCs [48,51,71,72]. Concomitantly, the remaining polyprotein comprising the glycoprotein precursor (E3-E2-6k-E1) is routed to the endoplasmic reticulum (ER). The N-terminal portion of the E3 glycoprotein contains a signal peptide which interacts with the ER membrane, translocating the polyprotein to the ER lumen. Once in the ER, the host signalase recognize both the N and C-terminal portions of the 6k, creating the products E3E2, 6k, and E1, which will be anchored in the organelle membrane. Next, E3E2 and E1 associate to form dimers that move with 6K through vesicles to the plasma membrane. Finally, E3E2 is cleaved by furin proteases present in a post-Golgi compartment, generating the glycoproteins E3 and E2 (Figure 2) [40,44,71,72].

Notably, the 6k protein is palmitoylated, enabling it to anchor to the membranes. This protein is involved in the transport of E1 and E2 to the plasma membrane and is a viroporin that enhances membrane permeability in infected cells by forming ion channels [42]. The 6K protein presents two transmembrane helices, classified as a Class II subclass A of viroporin types, connected by a short cytoplasmic loop [42,73]. In addition, the 6k gene undergoes a frameshift event and starts to produce the TF protein [42]. Within the gene encoding 6K, a heptanucleotide slip site (UUUUUUA) is highly conserved among the alphaviruses that causes a -1 ribosomal shift. This frameshift produces the TF protein in place of 6K and E1 with a frequency ranging from 5% to 40%, depending on the alphavirus species [74,75]. The 6K and TF proteins have an identical N-terminal portion (including the first transmembrane domain with ion channel activity) but differ in the C-terminal portion. The TF protein contains a C-terminal extension of the 6K protein and a longer cytoplasmic domain than 6K [74]. Functionally, TF seems essential for viral assembly, presenting a virulence role as antagonist of the host type I interferon responses [76].

### 2.4. Exit

The complete intracellular cycle of viral replication takes place four hours after the virus enters the cell. Virus release occurs after the anchoring of the viral capsid in the cell membrane; the viral lipid envelope is then formed through the budding mechanism (Figure 2) [72,77,78].

NC interacts with E2 proteins, initiating the process of assembly and the budding of viral particles [79,80,81,82]. This interaction occurs in a pocket (hydrophobic on the bottom and hydrophilic on the top) in the C-terminal domain of protein C, to which the E2 protein is non-covalently linked through its C-terminal domain [41]. In this interface, a consensus motif containing threonine, proline, and tyrosine (TPY) is present, which is conserved within the genus *Alphavirus* [83]. Previous results have suggested the co-transport of C and E2 to the plasma membrane, as both proteins colocalize in mobile vesicles in the cell cytoplasm [79].

The viral envelope is enriched with cholesterol and sphingolipids precisely because, in mammals, it sprouts from host cell membrane domains rich in these components [49,84]. These sites are present both in the cell body and in intercellular extensions induced by the virus itself (a phenomenon called cell-to-cell transmission) that facilitates the infection of neighboring cells [85]. The alphavirus exit from cells is determined by host factors like actin cytoskeleton remodeling, which promotes the transport of glycoproteins to the plasma membrane. Microscopic studies have revealed actin rearrangements and the accumulation of actin clusters in the cytoplasm at the final steps of viral infection with the co-localization of E2 in these foci and along the filaments [86]. The precise mechanism(s) of cleavage of the budding particles is still undetermined. It is unknown whether only the NC’s interaction with the envelope is strong enough to detach itself; nevertheless, other viral proteins, like 6K and TF, are involved in the process [42,87]. Studies with mutants in the 6K and TF proteins have shown that they play an essential role in the viral assembly and budding, although the exact mechanism is not completely understood [42]. According to Ribeiro-Filho et al. (2021) [41] the cryo-EM structure suggests that MAYV capsid assembly is based on electrostatic contacts between the C proteins. In addition, data suggest a probable action of nsP1 by inhibiting tetherin, an interferon-induced membrane protein, which has the action of restricting the exit of enveloped viruses [88,89,90]. Additional studies are needed to define the precise budding mechanism of alphaviruses from infected cells.

### 2.5. “Cell-to-Cell” Transmission

Alphavirus infection induces drastic remodeling of the cell cytoskeleton. This remodeling promotes the production of two types of cell extensions, the short- and long-filopodia-like structures [85,91] with length, contacts, and components at their main differences. The short ones are from 2 to 7 µm in length and comprise only F-actin filaments. Its induction requires the E2-C interaction [84,92] and is involved with nsP1, as assays performed using only structural proteins did not trigger such formations, and specific viral protein labeling demonstrated the co-localization of E2-C and nsP1. However, it is unclear whether nsP1 can mediate cell-to-cell transmission or act to protect viral particles from neutralizing antibodies [79,91].

The long intercellular extensions have more than 10 µm, can reach up to 60 µm, and are composed of F-actin and tubulin [79,91,93]. They are usually identified when there is a production of active structural proteins, with E2, E1, and C detected throughout the length of the filament. These extensions emerge from an infected cell towards to an uninfected cell through retrograde actin transport, and during this process viral particles moved and budded from the infected cell through these extensions. The contact tip of the filopodium-like structure is usually flat and does not fuse with the other cell, hence the name “virological synapse” [94]. Interestingly, there is a loss of stress fibers in infected cells in the late periods of infection, correlating with the appearance of extensions [79,91].

The “cell-to-cell” transmission requires the budding and release of active viral particles. Thus, transmission is not mediated by viral RNA transfer, replication complexes, or type II cytopathic vacuoles (CPVIIs) [86]. This is in agreement with the observed lack of cytoplasmic or membrane continuity between the infected cells and the target [93]. Intercellular transmission prevents the action of neutralizing antibodies on the virus, suggesting that nascent particles are somehow protected from the extracellular environment [85,95,96]. The data also strongly suggest that infection of target cells occurs by endocytosis and fusion triggered by low pH, as does initial infection. Furthermore, the host determinants for such a phenomenon to occur are unknown, and such structures are species and/or cell-line dependent.

## 3. Antivirals

Presently, there are no specific antiviral treatments or vaccines against MAYV infections. In urban environments of endemic areas, vector control through insecticide spraying is the only available measure to prevent infection. The clinical management of infected patients is limited to drug administration such as analgesics, non-steroidal anti-inflammatory drugs, and antipyretics to relieve symptoms.

Several natural and synthetic compounds have been evaluated to identify those with low cytotoxicity to host cells and the compounds’ ability to inhibit viral infection [97]. The parameters most commonly used for this purpose are the cytotoxic concentration for 50% of the cells (CC50), inhibitory concentration for 50% of the infected cells (IC50), and the selectivity index (SI) [98,99]. The SI is a factor calculated using the two previous parameters (CC50 and IC50), facilitating the comparison of compounds in terms of cytotoxicity and antiviral potency [100]. Substances that present antiviral activity are able to inhibit critical steps of viral replication, like events related to virus entry into host cells, activity of enzymes involved in the replication complex (proteases, viral helicase, and RNA-dependent RNA polymerase), or in late stages during particles assembly and release to the extracellular medium [38].

In addition to substances that directly affect the activity of virus proteins or enzymes, some compounds can modulate the host response and affect virus replication. Such substances can present the advantage of broad-spectrum use against other related alphaviruses. However, these compounds must be carefully studied, as the side effects generated by their use can cause disorders compared to compounds that act directly on virus proteins or enzymes [101]. In this review, we describe the state-of-the-art on promising substances/molecules with anti-MAYV activity, which are demonstrated in Figure 3.

### 3.1. Virucidal Compounds

Compounds or plant extracts are considered virucidal when they interact directly with the viral particle and, consequently, prevent infection onset. Several reports indicate that the flavonoid proanthocyanidin ([(-) epicatechin-(4β -> 8)-(-)-4′-methylepigallocatechin]; PAC) isolated from methanol extraction of *Maytenus imbricata* (*Celastraceae*) roots showed a virucidal effect on MAYV. This compound acted directly in MAYV particles and not in the host cells, with an EC50 (effective concentration for 50% of infected cells) of 37.9 ± 2.4 µM and a SI above 43 [102]. Ginkgolic acid, a substance isolated from the *Ginkgo biloba* plant, has a similar effect on MAYV infectivity [103].

Hydrophobic molecules like Co-protoporphyrin IX and Sn-protoporphyrin IX also have virucidal activity. By elucidating the mechanism of action of these compounds, it was found that they act on the conformation of envelope proteins, deforming the viral particle and preventing adsorption of receptors on the cell plasma membrane in an in vitro model [104].

Compounds present in the extract of *Schinus terebinthifolius* fruit, which is a condiment widely used in cooking, were evaluated against MAYV infection and proved to be virucidal in an in vitro model. Four substances (two bioflavonoids and two ethyl acetate) were isolated, and three of them had SI (60, 12, 98, 6, respectively) greater than that of ribavirin (used as antiviral control; SI = 8) [105]. The virucidal mechanism of these substances remains unknown. Another isolated compound is punicalagin from the extract of pomegranate (*Punica granatum*). Both extract and isolate proved virucidal for MAYV, with SI = 14 and 48, respectively [106]. The mechanisms of action for MAYV inhibition are still unclear. Studies using SARS-CoV2 demonstrated that its action is through the reversible and non-competitive binding to the viral surface glycoproteins [107].

### 3.2. Compounds That Interfere with Adsorption and Internalization of the Virus Particle

Some substances/molecules can prevent the anchorage, entry, and internalization of MAYV to the cell by inhibiting fundamental pathways for the initiation of viral replication. For example, it was found that extracts from *Cassia australis* leaves have an anti-MAYV effect, especially those containing condensed tannins (SI = 33; relative potency—RP = 16.5). This substance seems responsible for the antiviral activity since, characteristically, tannins have the ability to produce complex lipo and glycoproteins. The viral envelope is mostly composed of glycoproteins, so the compound possibly inhibits viral adsorption and internalization [108].

Bovine lactoferrin, a multifunctional globular protein isolated on a large scale from bovine milk and easily found commercially, acts by blocking the entry of MAYV into the host cell. The block seems to depend on the presence of sulfated glycosaminoglycans on the cell membrane surface, preventing viral anchorage [109].

Compounds already employed in the clinic for the treatment of other conditions or diseases, such as suramin, used to treat African sleeping sickness and onchocerciasis, also showed anti-MAYV efficacy. After treating cells with this drug, the inhibition of MAYV entry was evidenced, possibly because it causes conformational changes in the viral envelope proteins, impairing fusion with the host cell membrane [110]. A study analyzing the in vitro infection with CHIKV suggested that suramin interacts with the E2 protein, inhibiting the adsorption and/or conformational changes necessary for membrane fusion [111].

Another class of drugs is that based on antibodies. Only a few studies characterizing the cell receptors involved in MAYV entry into host cells are available [46,51,112,113,114]. As mentioned previously, it has been recently shown that the Mxra8 receptor is the putative entry molecule for MAYV and other arthritogenic alphaviruses and, thus, a potential therapeutic target. Using the Fc-Mxra8 fusion protein or anti-Mxra8 monoclonal antibody as treatment, in vitro infection by MAYV, CHIKV, Ross River virus, O’nyong nyong virus (ONNV), and Barmah Forest virus was reduced. Additionally, mice (C57BL/6) infected with CHIKV and ONNV and treated with Fc-Mxra8 fusion protein or anti-Mxra8 blocking antibodies had reduced infection and disease signals (foot swelling) [50].

Specific anti-MAYV neutralizing antibodies are also available, representing potential therapeutic targets to interfere with viral adsorption and internalization. Of the 151 hybridomas generated in the study by Earnest, J.T. et al. (2019) [115], 11 presented neutralization activity against MAYV. Neutralizing antibodies bind to E2 and E1 protein from D and L genotypes, with EC50 lower than 10 ng/mL. Additionally, the anti-MAYV neutralizing antibodies presented cross-reactivity and also inhibited the in vitro infection by other alphaviruses, like UNAV and CHIKV [115]. The neutralization of MAYV in vitro infection occurred mainly through monoclonal antibody recognition of the E2 protein, as previously demonstrated for antibodies produced against CHIKV [116] and Eastern Equine Encephalitis virus (EEEV) [117]. It was also demonstrated that an antibody cocktail that recognizes E1 and E2 had a greater effect in in vivo infection of C57BL/6J mice [115]. There are reports of humanized monoclonal antibodies that recognize different alphaviruses, like MAYV and CHIKV. These monoclonal antibodies were developed through expression of the E3-E2-E1 protein, using a more conserved region among the arthritogenic alphaviruses. Two monoclonal antibodies, DC2.M16 and DC2.M357, were generated and recognized the B domain of the E2 protein; they promoted neutralization, possibly through phagocytosis performed by monocytes (in vitro test) [20].

### 3.3. Compounds That Interfere with Replication, Morphogenesis, and Viral Exit

There are substances/molecules with inhibitory activity beyond viral entry, affecting steps such as viral replication, translation and the processing of viral proteins, morphogenesis/maturation, and viral sprouting, which are critical events for the formation of viral progeny [26].

Belonging to the class of immunomodulatory proteins, interferon alpha (IFN-α) is one of the viral inhibitors more widely used as an in vitro molecule to treat MAYV infection in cell lines [118,119,120,121]. Analysis by SDS-PAGE and protein densitometry showed that the inhibition of MAYV protein synthesis is heterogeneous after treatment with IFN-α, acting more significantly on glycoproteins E1 (99%) and E2 (79%) and their precursors (84% and 77%, respectively) compared to protein C (28%) [120]. This phenomenon can be explained by the increase in intracellular pH caused by treatment with IFN, compromising the traffic of proteins in the endoplasmic reticulum and Golgi complex, as previously shown in in vitro infections by vesicular stomatitis virus [122], herpes simplex viruses [119], CHIKV, ZIKV, and SARS-CoV-2 [118] and in the human treatment of hepatitis B [123].

Ribavirin (RIBV), another compound used in the clinic to treat viral infections [124,125,126,127,128] and widely used to control MAYV infection, has well-known pharmacological properties, such as bioavailability and toxicity. The putative mechanism of action of this compound is through (1) inhibiting enzymes related to the capping reaction of the viral RNA, since it is a guanosine analog; (2) suppressing the viral RNA synthesis by inhibiting the enzyme inosine-5′-monophosphate dehydrogenase; or (3) when phosphorylated, it can directly interfere with viral RNA synthesis during transcription [126,129,130]. RIBV has already been shown to be effective against CHIKV in mouse models, mainly when associated with doxycycline [131].

Furthermore, it has been demonstrated that compounds fractioned from plant extracts also impair MAYV replication. An example is quercetin isolated from *Bauhinia longifolia* (Bong.), which has a robust anti-MAYV activity (SI = 94; relative potency = 5) compared to ribavirin (SI = 8), and is used as a control compound [132]. Although the exact mechanism of action of quercetin during MAYV infection remains unclear, it has been suggested that it may be due to inhibition of viral RNA metabolism as observed for DENV-2 [132,133].

Epicatechin extracted from *Salacia crassifolia* is another compound that showed an anti-MAYV effect (SI = 7; RP = 2). There is strong evidence of the interaction of epicatechin with some viral components, possibly blocking important stages of the replication cycle and formation of new viral progenies without interfering with host cell metabolism. However, the precise mechanism of action of the compound action remains unknown [102].

In addition to natural compounds, synthetic molecules like thienopyridine derivatives also stand out as antivirals. Candidate compound 104 (SI = 125), for example, has low toxicity risks and severely influences viral protein synthesis, probably by inhibiting nsP2 transcription. There is a deficiency mainly in the assembly of the MAYV particle, favoring the production of non-infectious particles [134]. This compound showed promising action in inhibiting MAYV replication, acting at different points in the viral cycle. This conclusion was supported because (a) the viral protein synthesis was highly impaired; and (b) transmission electron microscopy after treatment with the candidate compound indicated action on the viral envelope [134].

Additionally, drugs already employed to treat other pathologies have shown anti-MAYV activity, such as ammonium chloride and chloroquine. Their action is demonstrated by a large amount of virus within intracytoplasmic vacuoles, in addition to free precursors and proteins (possibly capsid or nucleocapsid) in the cell cytoplasm, which are rarely seen during untreated infection. This effect possibly occurs because these substances raise the pH of the organelles of the Golgi complex, thus inhibiting the transport of glycoproteins to the membrane. In addition, this pH change can impair the function of some enzymes involved in the processing and maturation of viral glycoproteins, also affecting sprouting and viral infectivity. Consequently, these drugs seem to force the virus to follow an alternative morphogenesis route, leading to premature intracellular viral maturation [135].

The drugs EIDD-1931 and favipiravir have also been efficient in inhibiting MAYV infection. In the clinic, they are used to treat influenza A and B and resistant pandemic influenza viruses. Both drugs acted on MAYV viral replication in in vitro models by inhibiting polymerase and causing lethal mutagenesis during viral replication [110], similar to what has been reported for CHIKV [136,137] and Venezuelan Equine Encephalitis virus (VEEV) [138]. In a C57BL/6 mouse model, favipiravir pre-treatment reduced infection, RNA copies, and viral particle production in different tissues (e.g., brain, liver, heart, and spleen) during infection by MAYV [139], CHIKV [140] and Western equine encephalitis virus [141]. However, when the treatment of this compound is post-infection, of all tissues tested (e.g., brain, liver, heart and kidney) only the right quadriceps muscles showed a statistical decrease in MAYV infection [139]. A reduction in footpad swelling was also found in an animal model for the evaluation of arthritis and arthralgia triggered by MAYV [139]. In addition, in a retrospective study with pregnant women who used favipiravir, no major teratogenic effects were observed [142].

Inhibitors of the membrane-associated kinase family, such as Dasatinib and Torin 1, appear to act more specifically in viral translation without interfering with host mRNA translation [143,144]. It has been demonstrated that Dasatinib and Torin 1 inhibit the translation of subgenomic mRNA, resulting in a decrease in viral infection not only for MAYV but also for other alphaviruses [145].

In addition to drugs used as antivirals, some antibiotics have also been evaluated as potential anti-MAYV molecules. Monensin is an example of a natural antibiotic used as an additive in ruminant feed to increase feed efficiency. Its action, which has been previously described for other alphaviruses, such as the Semliki Forest and Sindbis viruses, has also been evaluated for MAYV. Treatment of both vertebrate and invertebrate cells with Monesin showed severe inhibition of MAYV replication, although viral morphogenesis was not affected [146].

### 3.4. Drugs That Modulate Host Response to Viral Infection

As previously mentioned, substances that may act on specific host factors are also promising alternatives for the development of broad-spectrum antiviral drugs. Nevertheless, there are still gaps in understanding the MAYV infection cycle in the host cells that would make the development and testing of this class of compounds a challenge.

After a high-throughput screening of 52,000 compounds, a small molecule named C11 that activates the signaling of the innate immune response was identified. C11 acts as an agonist of the adaptor protein STING and activates antiviral genes through phosphorylation of IRF3, mainly inducing the IFN type I response. Since alphaviruses are sensitive to the IFN response, including MAYV, this substance decreased the viral titer in an in vitro model [147].

Brefeldin A, a fungal metabolite reported to block newly synthesized endoplasmic reticulum protein transport, can also inhibit MAYV replication. By analyzing its effect during infection, a drastic reduction in the synthesis of viral proteins in mammalian cells was found which may or may not affect their glycosylation [148].

Prostaglandin A1 (PGA1) is another compound that has been shown to be effective against MAYV. This widely studied compound plays a role in several physiological and pathological processes, in addition to influencing the replication of several viruses like SINV [149], vaccinia virus [150] and Sendai virus [151]. After treating Vero cells with PGA1 and challenging them with MAYV, inhibition of E1 and E2 glycoprotein synthesis was found to be associated with triggering the synthesis of 70 kDa heat shock cell protein (HSP) [152].

The cyclic ketones, which are classified into different groups based on their chemical structure, have also been evaluated for their activity against different microorganisms and diseases. Among them, the xanthenodiones stand out as anti-bactericidal, anti-leishmania, antifungal, antitumor, anti-trypanosome, and antiviral substances [153,154]. Several compounds of this group were produced from the mixture of 1,3-diketone, aldehyde, and ZrOCl_2_8H_2_O and evaluated during in vitro infection by MAYV [155]. Only one of them, compound 9, remained in the test, as it obtained the best SI (15.8) [156]. However, when evaluating its anti-MAYV action, it was found to be effective only in pre-treatment or in high concentrations (338.8 µmol/L) [156], unlike what was verified for ZIKV, in which it interacts with viral envelope proteins preventing adsorption [155]. The likely mechanism of action is unclear, and it is believed that it may act on intracellular events. This hypothesis is based on some antiviral assays where there was no virucidal action or activity in the early stages of entry, as there is no decrease in infection compared to the positive control [134,156].

Drugs that act to inhibit the synthesis of new long-chain fatty acids, like orlistat (anti-obesity FDA approved) and cerulenin (antibiotic in test phase), which impair the activity of fatty acid synthase, as well as the substance CAY10566 that acts on the activity of stearoyl-CoA desaturase, have been shown to reduce in vitro MAYV replication. These enzymes seem important in the MAYV replication cycle and other arthritogenic alphaviruses, as they interfere with the plasma membrane anchoring of the nsP1 protein present in the replication complex [157]. Furthermore, cerulenin, considered an inhibitor of lipid synthesis, has been shown to reduce MAYV replication, probably preventing the incorporation of [3H] glycerol into lipids at any time after infection. Activity analysis at different time points during infection showed that the inhibition was more pronounced at the beginning of the infection, as it interferes with the synthesis of viral proteins [158].

Proteasome inhibitors are another class of virus inhibitory substances that can affect different stages of the replication cycle of CHIKV, rotavirus, and vaccinia virus [159,160,161]. Many viruses have evolved to hijack the proteasome machinery to make their replication more efficient, as this pathway is essential for cellular processes such as the cell cycle, immunity, and autophagy [162,163,164]. These drugs are described in the literature as acting at different stages of the viral replication cycle, both in protein expression [165], maturation [166] and budding [167,168]. For MAYV infection, the compounds MG132 and lactacystin have been shown to modulate the synthesis of E1 and nsP1, constituting critical steps during the replicative cycle. This drug activity reduces the amount of infectious viral particles produced in in vitro assays [77]. Another study with EEEV and VEEV compared the effectiveness of MG132 with another proteasome inhibitor, Bortezomib, which is FDA-approved to treat some forms of myeloma and lymphoma. In an in vitro model, there was a decrease in viral genomic copies, possibly due to interference by the ubiquitination of the viral capsid, preventing viral RNA from going into the cell’s cytoplasm [169]. Furthermore, MG132 also showed activity in the inhibition of additional targets, like certain lysosomal cysteine proteases, calpains, and cathepsins [170].

Another example is a plant-derived compound, silymarin, obtained from *Silybum marianum*. This substance acts on MAYV replication with a SI = 29.6, but the mechanism of action remains to be elucidated. One hypothesis that explains the anti-MAYV activity of silymarin in vitro is the modulation of cellular oxidative stress triggered by viral infection, decreasing the formation of ROS and, consequently, the levels of MDA (malondialdehyde) and protein carbonyl [171]. The efficacy of silymarin was assayed in a non-lethal model of disease in BALB/C mice [172,173]. In addition to the drastic decrease of viral infection in various organs (e.g., liver, spleen, brain) and serum, there was a control of the oxidant activity caused by the virus (carbonyl protein and MDA) and an increase of antioxidants (catalase, glutathione, and SOD activity). Thus, protection of the liver (ALT and AST activity) and other organs and the controlling of pro-inflammatory factors like IFN-*b*, TNF, IL-6, and IL-1B were also observed [28,173,174,175]. The in vivo (mouse model) as well as in vitro cell assays findings support the hypothesis that the modulation of MAYV-induced oxidative stress increases disease severity [176].

Other cellular factors that appear to be necessary for efficient replication of MAYV, at least in a cell model, are the mitogen-activated protein kinases (MAPKs) [177]. These proteins are involved in several cell cycle stages, like proliferation, growth, cytokine production, and stress response, controlling and regulating transduction signaling pathways [178]. It has been shown that the main pathway of MAPKs involved in MAYV replication is p38 since the addition of the p38 inhibitor (SB203580) leads to a severe reduction in infection and the production of viable particles [177]. Studies carried out with enteroviruses [179] and SARS-CoV-2 [144] found that this pathway interferes directly with the synthesis of viral proteins. When treating MAYV infection with two compounds that act on this pathway, NR-7h (which blocks several isoforms of p38) and losmapimod (a compound used in Phase 2 clinical study for the treatment of individuals with facioscapulohumeral muscular dystrophy, which blocks p38) there is a reduction in the expression of E1 [177].

Another family of protein kinases, the proviral integration sites of Moloney murine leukemia virus (PIMs), was evaluated during MAYV infection [180]. PIMs act in different cellular pathways, including migration, metabolism, and proliferation, and are closely linked to the prognosis of some cancers, such as colon, pancreatic and prostatic [181,182,183]. Studies using an in vitro model have shown that PIM 1 inhibitor 2 (isoform 1 only) and AZD1208 (pan-PIM kinase inhibitor for all PIM isoforms) show decreased rates of MAYV infection, viral progeny and protein synthesis of E1 and nsP1 [184]. This activity was verified during ZIKV and UNAV infections but not in CHIKV [185,186].

## 4. Conclusions

In this review, we describe compounds that are potentially effective against MAYV infection. They include synthetic molecules extracted from natural products as well as approved medicines already used to treat other diseases. Drug repurposing seems to be a promising strategy since the toxicity parameters have already been determined for use in humans, making the final approval by regulatory agencies faster than with other drugs. Still, non-commercial compounds (synthetic or extracted from plants) do not have safety test data, as most are in the preclinical development stages. Despite this, chemical engineering can benefit and improve molecules that have promising inhibitory results, both the synthetic ones and those isolated from natural extracts. Chemical engineering of molecules could lead to reduced toxicity, as well as high antiviral activity in drugs.

Considering that MAYV infection causes an acute and debilitating disease and the recent spread of this virus to broader regions in Brazil and South America [10,14,187,188,189], there is an urgent need for effective anti-MAYV compounds. Additionally, the lack of knowledge on the precise mechanism of MAYV replication and pathogenesis makes this neglected emergent virus a potential threat to all tropical and sub-tropical areas of the globe. Finally, further studies on MAYV replication and pathogenesis, including animal models, are essential for the future development of new, effective, and safe anti-MAYV drugs.

## Figures and Tables

**Figure 1 viruses-14-01787-f001:**
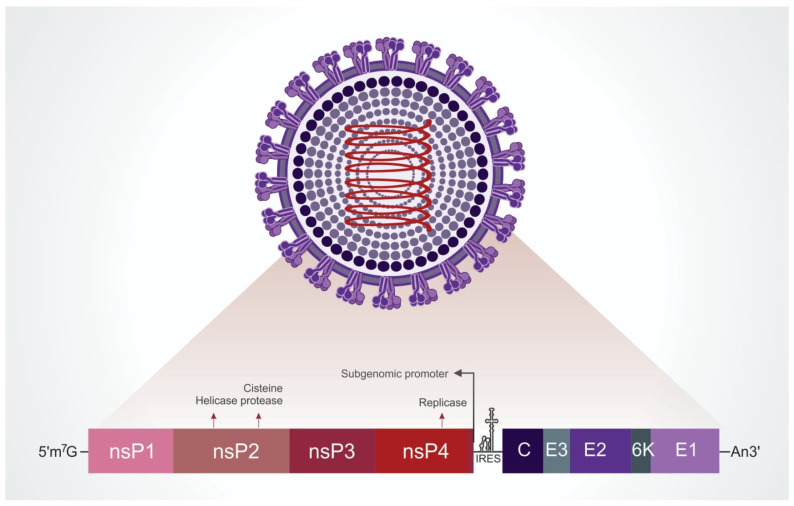
Schematic representation of MAYV particle and genomic organization. Electron microscopy data [41] shows that the viral particle is about 70 nm in diameter, constituted of structural proteins (in purple scale: C—capsid; E1—envelope glycoproteins 1; E2—envelope glycoproteins 2; E3—envelope glycoproteins 3; 6K—protein that associates E1 and E2). In addition to these, non-structural proteins 1 to 4 (in red scale: nsP 1–4) are synthesized in the early phase of the replication process.

**Figure 2 viruses-14-01787-f002:**
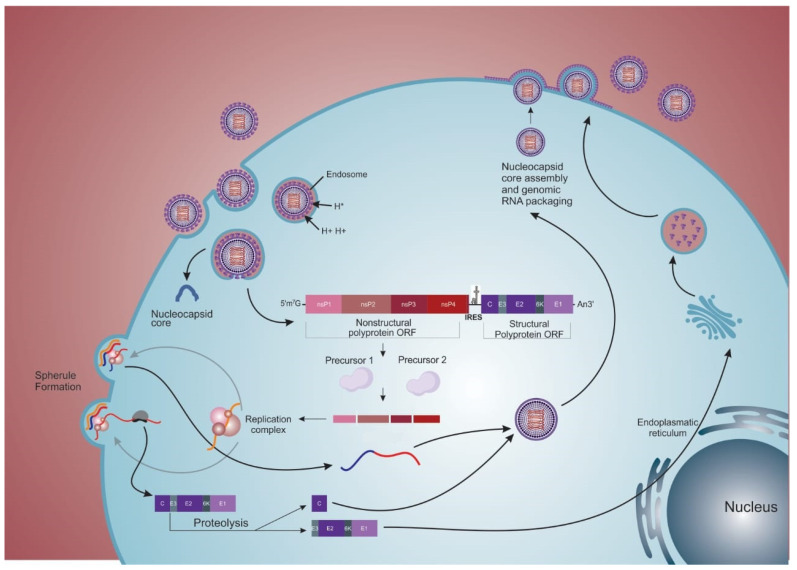
The replicative cycle of MAYV. Cell entry of MAYV is mediated by endocytosis and occurs via a clathrin-coated vesicle or, alternatively, caveolin. Internalization begins by binding the E2 virus with host cell receptors, possibly Mxra8. After endosome formation, acidification of its interior occurs, leading to structural changes in the viral envelope, exposing the E1 protein and promoting cell membrane and virus fusion. As a result, the nucleocapsid and viral genome are released into the cell’s cytoplasm. The genome is recognized by ribosomes, and a pair of nsP precursors is synthesized from the generated viral mRNA. The replication complex (RC) produced from the interaction of nsP is among the precursors. The RC catalyzes the synthesis of a negative-strand RNA that will, in turn, be the template for generating genomic (49S) and subgenomic (26S) RNA. The 26S structural polyprotein is generated, and the capsid protein is released and surrounds the 49S RNA through the action of the autoproteolytic serine protease, assembling the nucleocapsid. The remainder of the polyprotein is directed to the processing and maturation of E2 glycoproteins (through the E2 precursor protein; pE2) and E1 glycoproteins in the endoplasmic reticulum and Golgi complex. The mature glycoproteins will associate with and be transported to the cell membrane. The presence of the nucleocapsid in the membrane allows the recruitment of E1, initiating the process of viral assembly and the release of new MAYV and other alphavirus progenies through the cell membrane.

**Figure 3 viruses-14-01787-f003:**
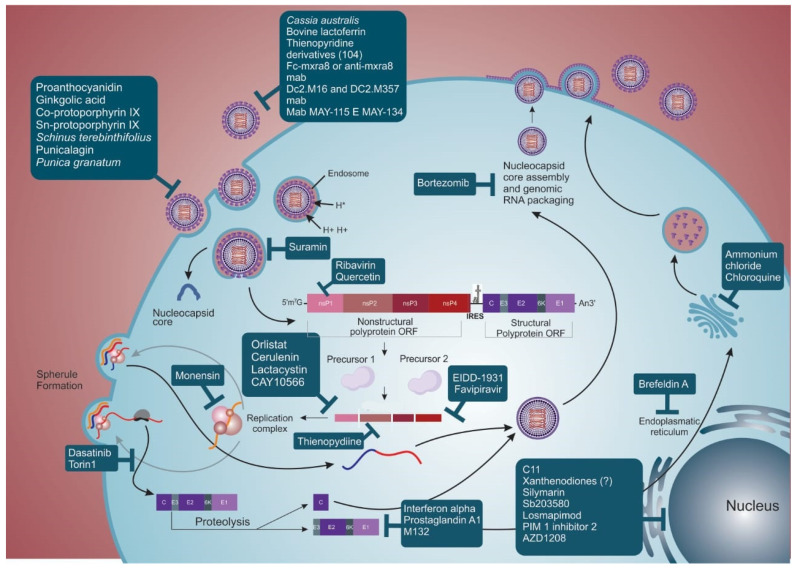
MAYV replication cycle and anti-MAYV compounds. Compounds with anti-MAYV activity are described at different time points of viral replication. Proanthocyanidin, ginkgolic acid, Co and Sn-protoporphyrin IX, schinus terebinthifolius, and punicalagin from *Punica granatum* have virucidal action. The compounds of *Cassia australis*, bovine lactoferrin, thienopyridine derivatives (no. 104), and the antibodies Fc-Mxra8 and anti-Mxra8, DC2M16 and DC2M357, MAY-115 and MAY-134 prevent adsorption of the viral particle. Suramin acts by blocking fusion during the viral internalization step. Ribavirin appears to interact with viral RNA before initiating replication, and quercetin acts to inhibit viral RNA metabolism. Some compounds interact with non-structural proteins, such as orlistat, cerulenin, CAY10566, and lactacystin that interact with nsP1; thienopyridine interacts with protease (nsP2); EIDD-1931 and favipiravir interact with nsP4. Monensin and epicatechin act during viral replication. Dasatinib and torin-1 inhibit the translation of subgenomic mRNA. M132 interferes during the synthesis of E1 and nsP1, possibly by compromising the action of lysosomal cysteine proteases. Interferon alpha inhibits the protein synthesis of envelope glycoproteins, possibly by increasing intracellular pH, compromising protein traffic. Prostaglandin A1 also interferes with the synthesis of envelope glycoproteins by acting on the synthesis of HSP70. Brefeldin A blocks the transport of envelope proteins in the endoplasmic reticulum. Ammonium chloride and chloroquine act on the Golgi complex, shifting the pH of the organelle and preventing the transport of glycoproteins to the cell membrane. Bortezomib interferes with the ubiquitination of the viral capsid, preventing the packaging of replicated viral RNA. Some compounds also target host factors like C11 (STING antagonist and IRF3 phosphorylation), xanthenodiones (intracellular event not determined), silymarin (oxidative stress modulation), SB203580 and losmapimod (p38 protein suppressors), PIM1 inhibitor 2 and AZD1208 (which interferes with the negative regulation of interferon-stimulated, like PML, OASL, and TRIM5 genes).

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
