# Peer review of "Mayaro Virus: The State-of-the-Art for Antiviral Drug Development"

_viruses, 2022, doi:10.3390/v14081787_

Round 1
Reviewer 1 Report
Andreolla, et al have reviewed Mayaro Virus Drugs and natural products. The review is very thorough and I only have a few minor corrections to recommend.
MINOR
1. Line 35- italicize the family names
2. Line 61 - name the mosquito that carries MAYV. Also, is there a particular suspected reservoir species?
3. Line 87 - "because it contains"
4. Fig. 1- "shows that the" and "constituted of structural"
5. Line 105- "are inferred from other alphaviruses."
6. Line 109 - I believe VEEV also has a receptor, maybe mention the other ones
7. Line 179 - can you mention where the attenuation sites in the 5' UTR are for LVS vaccine strains if there are any?
8. Line 190- some places say figure others say fig.
9. Line 260 - define 'pE2'
10. Lines 342-358- a separate section on antibodies with a header might be helpful
11. Line 364 - deleted 'the' interferon alpha
12. Line 385 - spell out 'Steud.'
13. Line 416 - I noticed that you hadn't mentioned where the virus has been found in vivo, may be state this earlier somewhere e.g. brain, liver, etc.
14. Line 417 - are these genotoxic?
15. Line 492 - capitalize Eastern
16. Is M132 an nsP protease inhibitor?
17. Line 574 - do you mean animal models?
18. Fig 3 - its not very clear as to which of these are polymerase inhibitors
Author Response
Dear Dr. Rafael Oliveira,
Thank you for sending us the reviewers' comments on our manuscript. We appreciate the largely positive reviews and address them below. The reviewer's comments are in bold and our answers in italic. We believe we have addressed all the reviewers' concerns
Kind Regards,
Claudia
- Line 35- italicize the family names
The sentence was modified as requested.
- Line 61 - name the mosquito that carries MAYV. Also, is there a particular suspected reservoir species?
The name of the potential urban vector (Aedes aegyti) for MAYV was included in the text.
- Line 87 - "because it contains"
The sentence was modified as requested.
- Fig. 1- "shows that the" and "constituted of structural"
The sentence was modified as requested.
- Line 105- "are inferred from other alphaviruses."
The sentence was modified as requested.
- Line 109 - I believe VEEV also has a receptor, maybe mention the other ones
The following expression was added throughout the text: "However, for encephalitic alphaviruses, such as VEEV and EEV, this receptor, when blocked, does not seems to affect the infection [51]."
- Line 179 - can you mention where the attenuation sites in the 5' UTR are for LVS vaccine strains if there are any?
Unfortunately, we didn't find the information on line 179. This line refers to the 6K protein and TF, thus not referring to the 5'UTR region. Nonetheless, there is a study describing the production and characterization of a live attenuated MAYV vaccine using the IRES region substitution approach. According to these data, the strain was successfully attenuated, with no viral replication in invertebrate (mosquito) or vertebrate cells. Furthermore, challenge with MAYV in a murine model showed protection in both immunocompetent and immunodeficient mice (Weise et al., 2014 doi: 10.1371/journal.pntd.0002969).
- Line 190- some places say figure others say fig.
The expression has been modified to figure only.
- Line 260 - define 'pE2'
The definition E2 glycoprotein precursor protein was included in the text.
- Lines 342-358- a separate section on antibodies with a header might be helpful.
We thank for the suggestion nevertheless, we believe that, as the structure of the article is focused on the replicative cycle of MAYV, together with some host factors that interact with viral replication, a specific topic of antibodies could disrupt the structural logic we created;
- Line 364 - deleted 'the' interferon alpha
The sentence was modified as requested.
- Line 385 - spell out 'Steud.'
This expression is not an acronym, it was a mistake and was removed from the text.
- Line 416 - I noticed that you hadn't mentioned where the virus has been found in vivo, may be state this earlier somewhere e.g. brain, liver, etc. DONE
The following sentences have been modified: "Both drugs acted on MAYV viral replication at in vitro model by inhibiting polymerase and causing lethal mutagenesis during viral replication [110], similar to what has been reported for CHIKV [136,137] and Venezuelan Equine Encephalitis virus (VEEV) [138]. In a C57BL/6 mouse model, favipiravir pre-treatment reduced infection, viral RNA copies, and viral particle production in different tissues (e.g. brain, liver, heart, and spleen) during infection by MAYV [139], CHIKV [140] and Western equine encephalitis virus [141]. However, when the treatment of this compound is post-infection, of all tissues tested (e.g. brain, liver, heart and kidney) only right quadriceps muscles showed a statistical decrease in MAYV infection [139]."
- Line 417 - are these genotoxic?
Studies demonstrate that they are not genotoxic and this information was added to the manuscript : "In addition, in a retrospective study with pregnant women who used favipiravir, no major teratogenic effects were observed [142]."
- Line 492 - capitalize Eastern
The virus name was corrected.
- Is M132 an nsP protease inhibitor?
The M132 molecule is not a protease inhibitor, but a proteasome inhibitor.
- Line 574 - do you mean animal models?
The information was added.
- Fig 3 - its not very clear as to which of these are polymerase inhibitors.
The nsP2 as the viral polymerase is described in the caption of figure 2.

Reviewer 2 Report
The manuscript entitled: “Mayaro virus: the state-of-art for antiviral drug development”, by Androella and collaborators, is an excellent revision on the subject. This manuscript is very well-written and show relevant and update information about drug discovery as potential anti-Mayaro molecules. In addition, this review is well-balanced and the diagrams showed in each figure are of a high quality. ¡Congratulation to the authors for the excellent work!
Author Response
Dear Dr. Rafael Oliveira,
Thank you for sending us the reviewers' comments on our manuscript. We appreciate the largely positive reviews and address them below. We believe we have addressed all the reviewers' concerns
Kind Regards,
Claudia